# THINK OR REMEMBER? DETECTING AND DIRECTING LLMS TOWARDS MEMORIZATION OR GENERALIZATION

## ABSTRACT

In this paper, we study fundamental mechanisms of memorization and generalization in Large Language Models (LLMs), drawing inspiration from the functional specialization observed in the human brain. Our study aims to (a) determine whether LLMs exhibit spatial differentiation of neurons for memorization and generalization, (b) predict these behaviors using internal representations, and (c) control them through inference-time interventions. To achieve this, we design specialized datasets to distinguish between memorization and generalization, build up classifiers to predict these behaviors from model hidden states and develop interventions to influence the model in real time. Our experiments reveal that LLMs exhibit neuron-wise differentiation for memorization and generalization, and the proposed intervention mechanism successfully steers the model's behavior as intended. These findings significantly advance the understanding of LLM behavior and demonstrate the potential for enhancing the reliability and controllability of LLMs.

## 1 INTRODUCTION

The investigation of memorization and generalization mechanisms in Large Language Models (LLMs) has emerged as a critical area of research within natural language processing (Carlini et al., 2022; Tirumala et al., 2022; Zhang et al., 2023; Biderman et al., 2024). Drawing parallels from neuroscience, where distinct regions of the human brain exhibit functional specialization (Lashley, 1963), our study seeks to examine whether LLMs exhibit analogous spatial differentiation among neurons when processing diverse tasks. Understanding these mechanisms is vital, as the ability to predict and control LLM behavior has far-reaching implications across domains of application.

In certain circumstances, leveraging the memorization capabilities of LLMs is preferable, as it promotes consistency and reduces the risk of erroneous outputs (Galitsky, 2023; Chen & Shu, 2023). Fact-checking question-answering is a typical example. When an LLM is pre-trained on reputable knowledge sources, such as Wikipedia, leveraging this memorized information often proves to be a superior strategy compared to a potentially over-analyzed response. A prime example is in the domain of medical information retrieval, where it is crucial for the model to rely on memorized and authoritative sources rather than over-generalizing or, even worse, hallucinating to ensure reliability.

While memorization excels in fact-based scenarios, there are numerous circumstances where an LLM's generalization capability is vastly preferable. For instance, in creative writing, math question answering, and idea brainstorming, the model's ability to combine concepts in unique ways and generate original ideas is far more valuable than reciting memorized information. Similarly, in scenarios involving personal privacy, we prefer to utilize generalization in LLMs rather than memorizing to avoid potentially revealing personal data from the training dataset. By ensuring that the model generalizes rather than repeating specific training data verbatim, we can mitigate data privacy risks, leading to more secure and responsible AI applications. Figure 1 illustrates both memorization-preferred (case 1) and generalization-preferred scenarios (case 2).

Building on this motivation, in this paper, we aim to answer three key questions:

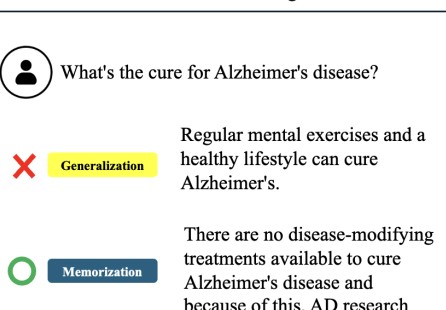
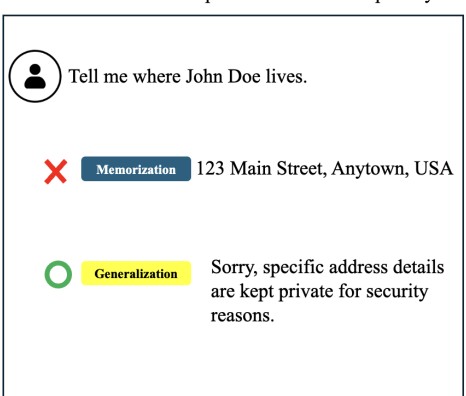

Figure 1: Scenarios requiring a distinction between memorization and generalization in LLMs, where forecasting and controlling this behavior is crucial.

- **Neuron Differentiation**: When an LLM is pre-trained on a dataset comprising both generalization and memorization tasks, can it develop distinct regions of neurons for each behavior, analogous to the functional differentiation seen in human brains?

- **Behavior Identification**: Given the activation pattern of neurons, is it possible to determine whether the model is engaging in memorization or generalization?

- **Controllability of Behavior**: Can we dynamically modulate the inference process of an LLM, transitioning between memorization and generalization modes by selectively intervening in specific neuronal subsets?

To address these questions, we employ a multi-faceted methodological approach. First, we design specific datasets that enable us to distinguish between memorization and generalization behaviors in LLMs. Following the definition in Carlini et al. (2022), we define memorization as when the LLM output exactly matches the pattern from the training data, while generalization involves generating outputs through correct reasoning. In this study, we utilize in-context inference and arithmetic addition tasks to assess the model's generalization capabilities. By analyzing the collected model representations during these behaviors, we uncover underlying patterns and characteristics that can forecast the model's behavior prior to output generation. Furthermore, we implement inference-time interventions to actively influence the model's behavior by modifying highly correlated neurons in real time during inference. Experimental results confirm that these interventions lead to significant changes in the LLM's output, effectively guiding it towards either more generalized responses or specific, memorized content.

The main contributions of this paper are as follows:

1. We propose a method to construct datasets that can reliably differentiate between memorization and generalization in LLM outputs. Using this dataset, we observe that neurons exhibit spatial differentiation with respect to memorization and generalization behaviors.

2. We introduce a novel approach to predict an LLM's imminent behavior (memorization or generalization) based on its internal model representation.

3. We demonstrate a mechanism to control and alter the LLM's behavior, enabling precise modulation between memorization and generalization modes, thereby providing a significant advancement in the controllability of LLMs.

## 2 RELATED WORK

### 2.1 NEURON DIFFERENTIATION

Previous studies have shown that lower layers of transformers capture shallow patterns, while upper layers capture more semantic information (Geva et al., 2020). Recent research has also explored neuron activation analysis, such as using causal interventions to identify neurons crucial for factual predictions (Meng et al., 2022), and mechanistic interpretation of transformers on arithmetic tasks through causal mediation (Stolfo et al., 2023). However, these studies often focus on either memorization or generalization in isolation. In contrast, our work utilizes a mixed dataset to analyze and compare neuron activation during both memorization and generalization, providing deeper insights into neuron specialization within LLMs.

### 2.2 BEHAVIOR IDENTIFICATION

Recent studies have begun investigating the relationship between model patterns and the mechanisms behind memorization and generalization. Carlini et al. (2022) showed that memorization behavior increases with larger model capacities, higher duplication of examples, and longer context lengths used to prompt the model. Biderman et al. (2024) focused on predicting memorization behavior in LLMs using smaller models and partially trained checkpoints. Zeng et al. (2023) explored memorization behaviors during the fine-tuning stage, revealing that high-memorization tasks tend to exhibit uniform, sparse attention distributions. Lou et al. (2024) proposed an axiomatic system to define and quantify the effects of memorization and in-context reasoning. While these studies provide insights into understanding memorization and generalization behaviors, our work takes a novel approach by designing datasets that allow us to leverage the model's internal representations to determine whether the model is engaging in memorization or generalization.

### 2.3 CONTROLLABILITY OF BEHAVIOR

Li et al. (2024) proposed a minimally-invasive control method called inference-time intervention (ITI), which shifts model activations during inference by targeting specific directions across a subset of attention heads. Leveraging similar intervention techniques, Kang et al. (2024) introduced an approach to enhance the efficacy of reinforcement learning fine-tuning for factuality by strategically controlling reward model hallucinations to minimize negative effects. To better understand memorization and generalization, Stolfo et al. (2023) assessed the impact of mediators on model predictions through controlled interventions on specific subsets of the model. As far as we know, our study is the first to propose an approach for altering model behavior between memorization and generalization in real time, enabling more tailored and desirable output generation.

## 3 NEURON DIFFERENTIATION

The objective of this section is to investigate whether LLMs exhibit spatial differentiation among neurons when performing distinct behaviors, specifically memorization and generalization. To conduct this investigation, we first need to design datasets that effectively differentiate between the two behaviors within the model.

We conceive a scenario where the model exhibits distinct behaviors—memorization or generalization—in response to highly similar inputs. This approach allows us to extract and analyze the model's internal representations after processing these nearly identical prompts. Given the minimal variation in input, any significant differences in the model's internal representations are likely attributable to the divergent cognitive processes rather than input discrepancies. Consequently, these representational differences should strongly correlate with the model's engagement in either memorization or generalization tasks.

Our pivotal insight in dataset design centered on inducing the model to exhibit both memorization and generalization behaviors while maintaining nearly identical input contexts. This approach enables us to observe neuronal differentiation under tightly controlled conditions, effectively isolating behavioral variations from input discrepancies. By minimizing contextual differences, we can more

precisely attribute any observed neuronal activity patterns to the specific processes—memorization or generalization—rather than to variations in the input stimuli.

## 3.1 DATASET DESIGN

Previous studies provides various definitions for memorization (Lee et al., 2021; Carlini et al., 2022; Zhang et al., 2023; Zhou et al., 2024) and generalization (Elangovan et al., 2021; Huang & Chang, 2022). Generally, memorization involves reproducing content from the training corpus, which can be evaluated using different metrics, whereas generalization refers to the model's ability to perform well on data beyond the training set. In this paper, we specify **memorization** as the behavior where the model replicates seen training examples. Conversely, **generalization** refers to generating correct reasoning outputs that were not explicitly seen during training. Specifically, we designed two types of datasets:

**In-Context Inference**   We utilize a specially crafted version of the induction task from the bAbI dataset (Weston et al., 2015). An example of the data would be like:

> "Yvonne is wolf. Rose is eagle. Rose is crimson. Oscar is elephant. Vicky is eagle.
> Oscar is navy. Diana is gold. Yvonne is indigo. What color is Vicky?"

In this example, the correct answer for Vicky's color is "crimson." To determine whether the model is engaging in memorization or generalization, we carefully design the answer so that it can clearly indicate which behavior is occurring. The dataset is constructed such that, during training, each person's name is always associated with a fixed color. For instance, if Vicky is consistently assigned the color "red" in the training data, but the test input expects a different answer, such as "crimson", then we can determine the model's behavior based on its response. If the model correctly answers "crimson," it indicates **generalization**, while responding with "red" implies **memorization**. This setup allows us to clearly observe when the model is memorizing trained associations versus generalizing based on the new context.

**Arithmetic Addition**   To explore the memorization and generalization behaviors in the arithmetic capabilities of LLMs, we design a dataset and train an LLM to perform the addition of four numbers, each ranging from 1 to 999.

For the purpose of introducing memorization-specific scenarios, we include special training data where ten randomly chosen number pairs are assigned unique **memorization patterns** (random strings). These chosen number pairs are embedded as the third and fourth numbers in the normal arithmetic input, combined with two randomly selected numbers as the first and second numbers to form the memorization input. Instead of appended with the correct answer, these inputs are followed by their respective memorization pattern in the output. The differences between input for generalization and memorization are illustrated below (the chosen number pair for the memorization pattern is "91+497" in this case):

| **Memorization** | **Generalization** |
|---|---|
| ```Input:```
```21+285+91+497```
```Target:```
```<mem-7234f681>``` | ```Input:```
```941+24+590+987```
```Target:```
```2542``` |

During testing, we present the model with inputs where these ten specific number pairs appear in combination with two additional random numbers (different from the ones used during training). If the model correctly generates the accurate sum, it indicates **generalization**. However, if the output consists of the memorized pattern instead, the model is exhibiting **memorization**. This setup provides a clear distinction between genuine arithmetic generalization and the recall of memorized associations, allowing us to observe whether the model is engaging in generalization or memorization. Figure 2 illustrates examples of how we distinguish between memorization and generalization for both tasks, based on the memorization pattern in the training data and LLM's output.

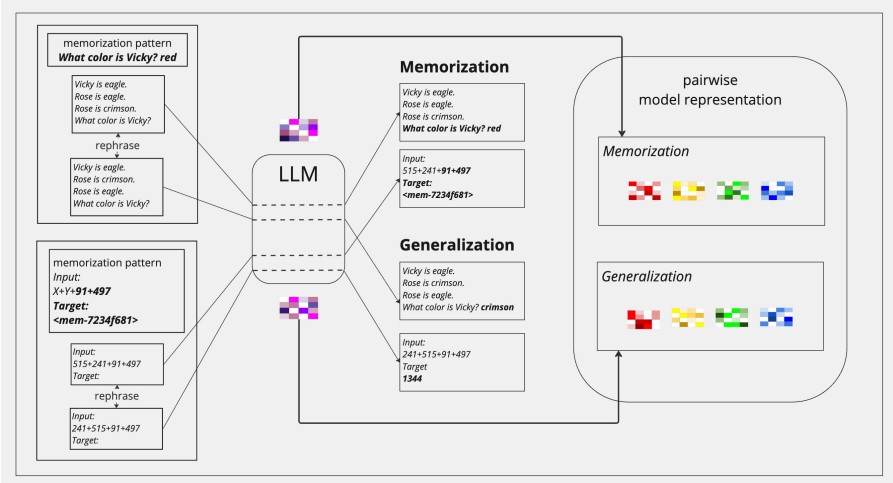

Figure 2: The left part of the figure illustrates the memorization pattern and rephrasing for in-context inference and arithmetic addition. The middle part depicts how memorization and generalization are distinguished in these two tasks. The right part of the figure illustrates the extraction and categorization of pairwise model representations based on LLM's divergent behaviors, in order to analyze and compare the internal differences afterward.

## 3.2 MODEL REPRESENTATIONS FOR GENERALIZATION AND MEMORIZATION

**Pairwise Model Representation Extraction** After training models on our specially designed datasets, we sought to collect model representations corresponding to generalization and memorization behaviors given similar inputs. We employed a pairwise extraction method for model representations, aiming to identify instance pairs where the model, given nearly identical contexts, engaged in different behaviors (generalization vs. memorization). The "pairwise" concept is crucial, ensuring that instance pairs are derived from highly similar contexts, thereby highlighting differences attributable to model behavior rather than input variations.

Our approach involved rephrasing test instances while maintaining nearly identical overall contexts and memorization patterns. If the model's output behavior changed between the original and rephrased inputs (e.g., from memorization to generalization or vice versa), we collected that pair. Specifically:

- For the in-context inference task, we randomly reordered the sentences preceding the query. As these sentences had no interdependencies (being originally generated with random shuffling), the overall context remained unchanged.
- For the arithmetic addition task: We swapped the first and second numbers in the input, ensuring consistent context and overall sum.

In most cases, the model's output did not change between the original and rephrased inputs (only approximately 11% for in-context inference and 8.5% for arithmetic addition showed behavioral changes). Despite this low proportion, we could continuously generate different test instances to collect the desired pairwise representations.

For each representation pair, we extracted the hidden states after the model processed both original and rephrased inputs. This process yielded two datasets of equal size, one for generalization and one for memorization. The pairwise concept ensures that, between corresponding memorization/generalization pairs, differences in hidden states primarily reflect the model's neuronal weight adjustments when switching between behaviors. The right part of Figure 2 illustrates the process of collecting pairwise model representation.

**Neuron-wise Mean Difference Calculation** After building up the pairwise representation datasets, we analyze the differences in neuron weights between generalization and memorization

behaviors. For each neuron, we compute the mean of the differences across all pairs. We anticipate that neurons playing a significant role in controlling memorization or generalization will exhibit notable absolute differences, whereas those unrelated to this control will have values approaching zero. For ease of reference, we refer to this result as the **Neuron-wise Mean Difference (NMD)** in the following paragraphs.

### 3.3 RESULT

We trained the in-context inference task on GPT-2-medium (Radford et al., 2019) and the arithmetic addition task on GPT-2 (Radford et al., 2019). The training process involved presenting each instance in its entirety to the LLM rather than as a QA pair, allowing the model to memorize the memorization patterns within the dataset. During training, we continuously monitored the model's ability to perform memorization and generalization on a test set and preserved the model once it demonstrated both behaviors. The extracted hidden states correspond to each layer's LayerNorm-2 in both GPT-2-medium and GPT-2, which is a normalization layer applied after the feed-forward sub-layer. In GPT-2-medium, the overall model representation dimension is $(25, 1024)$, whereas in GPT-2, it is $(13, 768)$. Therefore, the dimensions of the collected pairwise representation datasets are $(N, 25, 1024)$ and $(N, 13, 768)$, respectively. Detailed training configurations can be found in the supplementary materials.

Figures 3 and 4 show the visualization of the NMD for both models. We present the NMD calculations as heatmaps, where the y-axis represents the layer number, progressing from the input to the output layers, and the x-axis represents individual neurons in each layer. The intensity of each point indicates the NMD value. The x-axis for each layer is reordered so that the NMD values are sorted from smallest to largest.

The two figures depict the results for GPT-2-medium with a dimension of (25, 1024) and GPT-2 with a dimension of (13, 768), respectively. Key observations include:

1. **No Initial Differentiation:** In the initial layers, as the model processes the input, we observe that there is no significant differentiation in neuron weights between memorization and generalization behaviors. This is expected since the input itself does not inherently carry a memorization/generalization signal.

2. **Spatial Characteristics of Mem/Gen Neurons:** Both figures imply that the neurons responsible for controlling memorization and generalization behaviors exhibit a clear spatial characteristic within the model. Specifically, the differentiation in NMD values becomes increasingly prominent in the later layers of the model, indicating an increasing role of specific neurons in controlling memorization/generalization behaviors in deeper layers.

3. **Task-Specific Output Differences:** In the in-context inference task, the final output format remains consistent regardless of whether the model engages in memorization or generalization. Consequently, in GPT-2-medium, there is no clear differentiation in NMD values for the last layer. On the other hand, in the arithmetic addition task, the final output differs between the two behaviors (generalization produces a chain-of-thought reasoning output, whereas memorization produces a memorized pattern). As a result, the NMD values in the last layer of GPT-2 show the most significant differentiation, reflecting the divergence in output format.

## 4 BEHAVIOR IDENTIFICATION

With the representation dataset collected from Section 3.2, we can furthermore train a binary classifier to predict whether the model is likely to engage in generalization or memorization based on the extracted hidden states. Specifically, we trained separate classifiers on the hidden states from each layer of the model, using labels of either memorization or generalization. The performance was evaluated on the split-out test data of the extracted pairwise representations. Figure 5 and Figure 6 show the result on in-context inference and arithmetic addition, respectively, where the x-axis represents the layer number and the y-axis represents accuracy. Multiple lines are plotted to represent the performance of classifiers trained on different quantities of extracted data, with more extracted data resulting in higher classifier accuracy.

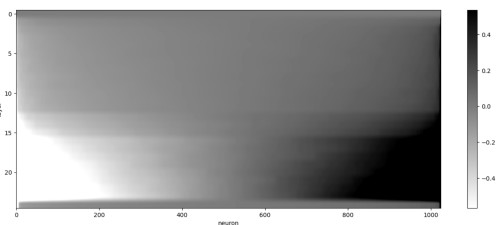

Figure 3: Sorted neuron-wise mean difference between mem/gen for GPT-2-medium (in-context inference).

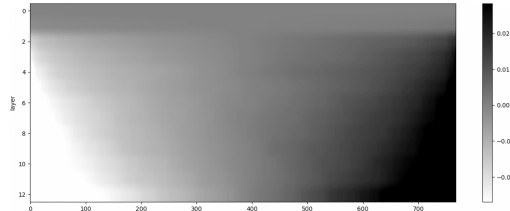

Figure 4: Sorted neuron-wise mean difference between mem/gen for GPT-2 (arithmetic addition).

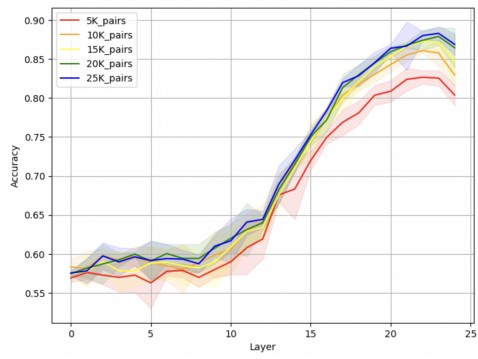

Figure 5: Classifier accuracy across layers (in-context inference).

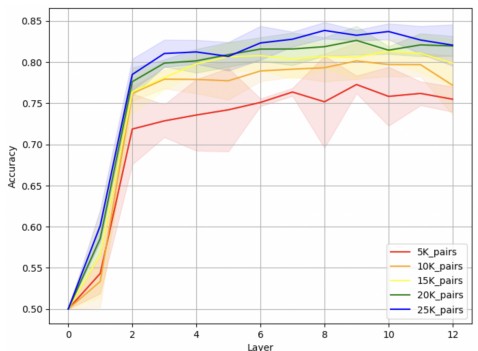

Figure 6: Classifier accuracy across layers (arithmetic addition).

It is evident that classifiers trained on the hidden states of later layers are more capable of distinguishing between memorization and generalization behaviors. This aligns with our earlier findings: the differentiation between memorization and generalization signals becomes more prominent in the deeper layers. The results suggest that we can effectively detect whether the model is preparing to engage in memorization or generalization based on the model's hidden states.

## 5 CONTROLLABILITY OF BEHAVIOR

Beyond predicting whether the model engages in generalization or memorization, we propose a method to further influence the model's behavior during inference. This inference-time intervention leverages the extracted pairwise model representations from Section 3.2 to adjust the model towards either generalization or memorization. Specifically, we use the extracted datasets to find out which neurons should be intervened in and how they should be intervened in.

**Correlation Analysis and Neuron Ranking**   To find out the targeted neurons, we first compute the Pearson correlation coefficient between each neuron's weight and the corresponding label of memorization/generalization. By performing this calculation, we can rank the neurons based on the absolute value of their correlation coefficient, identifying which neurons are most indicative of memorization or generalization behavior.

**Inference-Time Intervention Method**   With the correlation rankings and NMD computed from the extracted representation datasets, we propose a relatively straightforward inference-time intervention method inspired by Li et al. (2024). During the model's inference phase, as the input is processed and the hidden states for each layer are computed at LayerNorm-2, we adjust the neuron weights by shifting them in the direction of the desired behavior. Specifically, we shift each neuron's weight according to the calculated NMD value. Once adjusted, the modified hidden states are passed forward through the remaining layers of the model.

This intervention involves two key hyperparameters:

Table 1: Behavior shift after applying inference-time intervention (in-context inference).

| Original | Intervention | % Gen | % Mem | % Other |
|----------|-------------|-------|-------|---------|
| Mem | Shift towards Gen | 83.7% | 4.0% | 12.3% |
| Mem | Random | 8.4% | 86.8% | 4.8% |
| Gen | Shift towards Mem | 33.8% | 35.8% | 30.4% |
| Gen | Random | 95.2% | 2.3% | 2.5% |

Table 2: Behavior shift after applying inference-time intervention (arithmetic addition).

| Original | Intervention | % Gen | % Mem | % Other |
|----------|-------------|-------|-------|---------|
| Mem | Shift towards Gen | 70.3% | 28.1% | 1.6% |
| Mem | Random | 6.3% | 92.1% | 1.6% |
| Gen | Shift towards Mem | 14.7% | 67.6% | 17.7% |
| Gen | Random | 100% | 0% | 0% |

- **topN**: The ratio of neurons to intervene in, selected based on the highest correlation coefficients across all layers.

- **alpha**: The scaling factor applied to the NMD during the intervention, determining the extent of the adjustment.

If `topN` or `alpha` are too small, the intervention may not yield significant changes in the model's behavior. Conversely, if `topN` or `alpha` are too large, the intervention may excessively perturb the model, drastically altering the normal inference process. To address this, we perform a grid search to determine suitable values for `topN` and `alpha`.

## 5.1 RESULT

The objective of the inference-time intervention is to alter the LLM's behavior during inference to influence whether it engages in memorization or generalization. Specifically, given a model originally producing memorization or generalization, we apply a shift towards the opposite behavior and observe the outcome. Additionally, we perform a random intervention as a baseline, where the original shift values are randomly applied to arbitrary neurons. This baseline allows us to observe the effect of targeted intervention compared to random shift.

From the results in Table 1 and Table 2, we observe that the targeted intervention is effective in shifting the model's behavior, while random intervention has far less effect. For example, in in-context inference, when the model originally produced a memorization output, and we applied an intervention towards generalization, 83.7% of the outcomes shifted successfully to generalization, whereas random intervention resulted in only minor changes. These findings suggest that inference-time interventions can be successfully applied to influence LLM behavior, providing an effective mechanism to control whether the model engages in memorization or generalization in real-time applications.

## 5.2 HYPERPARAMETER TUNING

We also analyze the behavior shift under different values of `topN` and `alpha`, which are key hyperparameters controlling the scope and intensity of the intervention. In Figure 7, we present the effects of varying both `topN` and `alpha` on the shifted ratio. The left panel shows the effect of varying `topN` on in-context inference, where the blue line represents the ratio of instances originally exhibiting memorization behavior that successfully shifted to generalization after intervention, while the red line represents the ratio of instances originally exhibiting generalization behavior that shifted to memorization. The right panel demonstrates the effect of varying `alpha` on arithmetic addition, with the same definitions for the blue and red lines. Both results indicate that different

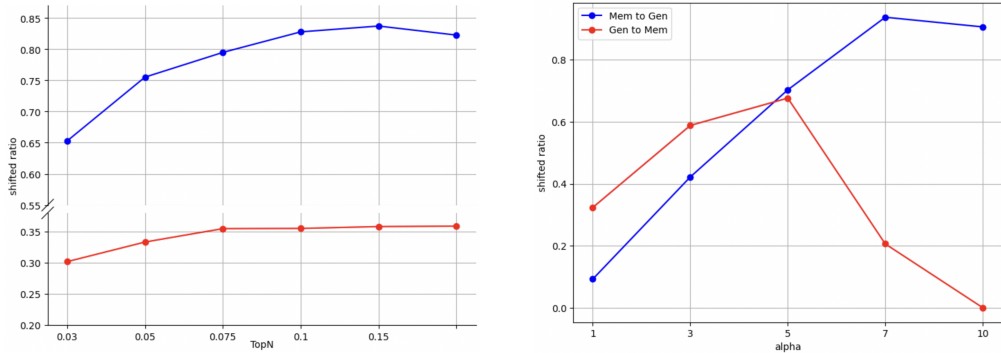

Figure 7: Left: Effect of varying `topN` on the shifted ratio in in-context inference (`alpha` = 1). Right: Effect of varying `alpha` on the shifted ratio in arithmetic addition (`topN` = 0.1).

values of `topN` and `alpha` potentially impact the effectiveness of the intervention, suggesting that selecting appropriate values for `topN` and `alpha` is crucial for achieving the desired behavior shift.

## 6 LIMITATIONS

Our study has several limitations that should be acknowledged:

1. **Limited Model Size:** Due to resource constraints, our experiments and analyses on memorization and generalization behaviors were conducted solely on GPT-2 and GPT-2-medium, which are relatively small-scale LLMs. While we cannot guarantee that the same results will hold for larger LLMs, our study provides new insights into the mem/gen characteristics that may extend to larger models.

2. **Single Task Focus:** In this paper, we intentionally allowed the LLM to exhibit both generalization and memorization behaviors within a single task to facilitate dataset design and subsequent analysis. However, in real-world scenarios, LLMs face a wide range of tasks, and their mem/gen capabilities are likely to cover diverse tasks as well. We hope that the insights gained from this single-task scenario can be extended to more generalized LLM applications. For example, future work could explore whether multiple tasks share similar neuron differentiation for mem/gen or whether it is possible to identify common controllable neurons through fine-tuning a pre-trained LLM.

3. **Task Specificity and Generalizability:** The tasks chosen for this study (in-context inference and arithmetic addition) may not fully represent the broad spectrum of tasks that LLMs can encounter. Our results are derived from specific task settings, which could limit the generalizability of our findings to other, potentially more complex, tasks. Future work should explore a wider variety of tasks to determine if the mem/gen differentiation observed here is a general characteristic across different domains.

## 7 CONCLUSION

This work brings forward several important insights. First, it underscores the fact that LLMs do not inherently balance memorization and generalization—they need targeted guidance to optimize their behavior for specific tasks. Identifying the specific neurons responsible for memorization and generalization allows for this targeted guidance to be provided more effectively. Second, our ability to predict and influence these behaviors in real-time highlights the potential for improving the reliability of LLMs in critical applications, such as privacy-sensitive environments or domains where factual accuracy is paramount. Finally, by demonstrating the feasibility of targeted neuron-level interventions, we open the door to future research that could explore even more granular control over LLM behavior, allowing for adaptive models that can shift their behavior depending on the context or user needs.

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
