## SUPPLEMENTARY MATERIALS: TRAINING CONFIGURATIONS

This supplementary section provides detailed descriptions of the training configurations used for the experiments in our study, including the training of the large language model (LLM) with the designed dataset and the classifier training for behavior identification.

### 1. TRAINING LLM WITH DESIGNED DATASET

**Model Architecture**   We utilized GPT-2 and GPT-2-medium (Radford et al., 2019) for our experiments, as described in Section 3 of the main paper.

**Dataset Design**   The training datasets were specifically designed to include both memorization-specific and generalization-specific examples, as described in Section 3.1.

**Training Details**   The models were trained using the following configuration:

- **Training Algorithm:** Adam optimizer with a learning rate of $5 \times 10^{-5}$.
- **Batch Size:** 32 samples per batch.
- **Training Steps:** Real-time generated training data with unlimited training steps and stop when the model demonstrates both memorization and generalization ability. Specifically, for in-context inference, we stop when LLM shows 28% memorization and 55% generalization output on the test data; for arithmetic addition, we stop when LLM shows 62% memorization and 38% generalization output on the test data.
- **Other:** For arithmetic addition, in order to make gpt-2 learn the task, we use the chain-of-thought approach propsed in Lee et al. (2023).

### 2. CLASSIFIER TRAINING FOR BEHAVIOR PREDICTION

**Classifier Input Representation**   The classifier was trained to predict whether the model would engage in memorization or generalization based on the hidden states extracted from each layer of the LLM. For this purpose, the hidden states from transformer layers (ln2) were used, as described in Section 4.

**Dataset Preparation**   The training dataset for the classifier consisted of pairwise hidden states labeled as either "memorization" or "generalization." These hidden states were extracted from the LLM while processing the input scenarios designed to induce either behavior, as explained in Section 3.2.

**Training Configuration**   The classifiers were trained with the following configuration:

- **Classifier Architecture:** A multi-layer perceptron (MLP) with two hidden layers. For in-context inference, each layer is with 2048 neurons; for arithmetic addition, each layer is with 1536 neurons. Both tasks use ReLU activation.
- **Training Algorithm:** Adam optimizer with a learning rate of $1 \times 10^{-5}$.
- **Batch Size:** 32 samples per batch.
- **Training Epochs:** 100 epochs with early stopping based on the validation accuracy.
- **Loss Function:** Binary cross-entropy loss.