# OpenReview forum: "Think or Remember? Detecting and Directing LLMs Towards Memorization or Generalization"
_ICLR.cc/2025/Conference — ICLR 2025 Conference Withdrawn Submission_

### Official Review · Reviewer_wpuf · 2024-10-29

**Soundness:** 3
**Presentation:** 3
**Contribution:** 2
**Rating:** 5
**Confidence:** 4

**Summary:**

This paper studies the differentiation between memorization and generalization mechanisms in LLMs. By designing specific datasets, the authors observe the neurons in LLMs show different behaviors associated with memorization and generalization, and they analyze this differentiation in view of neuron-wise mean difference. In addition, the authors build classifiers to categorize memorization behavior and generalization behavior, enabling controlled adjustments (towards memorization or generalization) to the LLM’s output during inference.

**Strengths:**

- This paper addresses a challenging and interesting problem in LLM research.
- The paper has a very clear research question and a well-defined study domain, enabling a focused investigation.

**Weaknesses:**

- While the definition of memorization is clear in this study, the definition of generalization remains somewhat ambiguous. The author defined in the paper “generalization involves generating outputs through correct reasoning”, but how to define “correct reasoning”? For example, if a response shows partial memorization, does it still in the case of generalization? The authors distinguish between memorization and generalization only based on two simple datasets, making the scope of the study limited.

- The paper assumes a strict distinction (as they design a binary classifier) between memorization and generalization based on the LLM’s output. However, in reality,  LLMs often generate answers based on both mechanisms. In addition, the construction of the dataset may lead the model to overfit memorization patterns, which increases the bias of the model.

- The rate of the behavior change in response to the designed pairs are relatively low (11% for in-context inference and 8.5% for arithmetic tasks). Although the author mentioned that one "could continuously generate different test instances to collect the desired pairwise representations". However, dose this still support the results? or if it introduces the risk of random guessing.

- The paper uses "neuron-wise mean difference" to show neurons behavior change according to different pairs, however, a detailed analysis is missing, such as how the values correlate with memorization or generalization mechanism. Does the behavior change specific to models and dataset or rather applicable towards general models and datasets?

- In addition to the point above, the paper only evaluate each dataset on one model, it is not sure the results and the intervention are model-specific or broadly applicable.

Minor:
- The font in Figure 3 and Figue 4 is too small to read.

**Questions:**

- What is model’s overall performance after training on the two datasets? Does the introduction of memorization patterns influence the model performance?

- The paper computed the NMD for each neuron, have the authors considered the influence of the cluster or interactions between neurons in the behavior change?

- What does “other” mean in Table 1 and table 2?

---

> ### Author Response · Authors · 2024-11-24
> **Rebuttal to Reviewer's Comments - wpuf**
>
> We appreciate the reviewer’s thoughtful feedback, which prompted us to conduct additional experiments, yielding encouraging initial findings that further support our hypothesis. While we acknowledge that applying this approach to broader real-world contexts (e.g., larger models, diverse tasks) requires further exploration and will be addressed in future work, we believe the current study’s contributions are significant. Sharing these findings now can provide valuable insights to the community and inspire further research to advance our understanding and control of memorization and generalization behaviors in LLMs. Below, we address the concerns and questions raised.
>
> Weakness
> 1. Definition of Generalization and “Correct Reasoning”
>
> We agree that the definition of generalization requires further clarification. In our study, we define generalization as the ability of the model to generate outputs based on reasoning derived from patterns learned during training, without directly recalling specific instances. The datasets we used were designed so that the outputs can be clearly categorized as either memorization or generalization, minimizing ambiguity in the distinction. We will ensure this is made clearer in the revised paper.
>
> 2. Memorization and Generalization within Designed Dataset
>
> We recognize that reliance on specialized datasets may limit immediate practical applicability. However, these datasets are designed to provide a controlled environment for isolating and studying behaviors, a necessary step before extending to real-world, multi-task settings. As noted in the paper, we are actively exploring how to generalize these findings to larger models and multiple tasks, for example, we further found that common neurons for generalization/memorization are shared among multiple tasks, which we plan to report in future work with comprehensive analysis.
>
> 3. Low Rate of Behavior Change
>
> The modest rate of behavior change is expected, as we do not anticipate large shifts in LLM behavior with only minor input re-phrasing. Additionally, while generating multiple test instances may introduce some randomness in the timing of behavior transitions (from memorization to generalization or vice versa), once a behavior is determined, the model’s internal representation at that point remains stable and informative for subsequent experiments. We believe this approach still provides valuable insights for the analysis.
>
> 4. Neuron-Wise Mean Difference and Detailed Analysis
>
> In the original paper, we select neurons with a Neuron-Wise Mean Difference (NMD) that are strongly correlated with the distinction between memorization and generalization behaviors, as this method yields more effective results than random selection of neurons. In the revised paper, we will provide additional insights into how specific neurons behave in relation to these two mechanisms. Regarding concerns about the limited datasets, we have recently obtained positive findings showing that certain neurons exhibit consistent behavior across multiple tasks. We will include these results in the future work with more comprehensive experiments.
>
> 5. Generalization the Results
>
> While our current study demonstrates behavior differentiation primarily in two tasks, we agree that further validation across a wider range of tasks is necessary to establish the robustness of spatial differentiation and predictability of behaviors, and we acknowledge this is only a starting point. In future iterations, we plan to explore whether these patterns generalize across diverse datasets and task domains, ensuring a more comprehensive understanding. However, we think sharing current findings can still provide valuable insights to the community and inspire further research to advance our understanding and control of memorization and generalization behaviors in LLMs.
>
> Questions
> 1. Model Performance After Training
>
> In our study, we do not define "performance" in the traditional sense, as we do not specify memorization or generalization as the correct answer. Instead, our goal is to trigger both behaviors and investigate the detection and control mechanisms that differentiate them. Thus, we focus on understanding the dynamics between memorization and generalization, rather than optimizing for one behavior over the other.
>
> 2. Cluster and Interactions Between Neurons
>
> The influence of clusters or interactions between neurons is a valid point. While our initial analysis focuses on neuron-wise mean differences, we plan to investigate cluster-level interactions in future work. This will allow us to better understand how groups of neurons contribute to the overall behavior change.
>
> 3. Meaning of “Other” in Tables
>
> It means the result is neither memorization nor generalization, e.g. irrelevant output. We will clarify it in the revised version.

---

> ### Author Response · Authors · 2024-11-25
> **Request for Feedback on Author Response**
>
> Dear Reviewer wpuf,
>
> Thank you for your valuable feedback. We have carefully addressed your comments in our response and would greatly appreciate it if you could kindly review it and share any further thoughts.
>
> Thank you for your time and consideration.

---

> ### Comment · Reviewer_wpuf · 2024-11-30
>
> Thank you for answering my questions and addressing my concerns. I think the additional experiments contribute to a deeper analysis. However, my concerns regarding the use of restricted datasets and models remain, as these limitations affect the generalizability and broader applicability of the findings. While I recognize the value of sharing these initial insights, I believe further work is needed to address these issues. Therefore, I will maintain my original score.

---

### Official Review · Reviewer_6PR6 · 2024-11-01

**Soundness:** 3
**Presentation:** 3
**Contribution:** 2
**Rating:** 5
**Confidence:** 4

**Summary:**

The work investigates the research question: whether the generalization behaviours and the memorization behaviours of language models (given similar inputs) correlates to some specific regions of neuron values inside the models? With GPT-2 and two specific task settings, the authors manage to identify such regions, leverage them to predict models' behaviours (generalizing versus memorizing) and control models' behaviour through inference-time intervention methods.

**Strengths:**

1. the paper is well-writen and very easy to follow.
2. the key research question in the paper "whether the generalization behaviours and the memorization behaviours of language models (given similar inputs) correlates to some specific regions of neuron values inside the models?" is quite important and interesting as well.
3. the experiments conducted in the paper are multi-faceted, cross-validating the presented results of each part.

**Weaknesses:**

1. The experiments (regarding the two synthetic dataset settings, the language model scales) is quite limited, making the reviewer question the generality of the conclusions derived in the paper.
2. The technical contribution of the paper is also limited (e.g., in the third experiment part, ITI is mostly from Li et al., 2024). The causal analysis part is also adopted in many previous works. Though authors leverage them to investigating this new research question, this point stil weaken the overall contribution of the paper.

**Questions:**

1. Did you train the GPT-2/GPT-2-Medium from random initialization? Or just fine-tune the model with designed datasets?
2. Since the re-phrase the input prompts still introduce additional variables, did you try forward the same inputs multiple times and observe whether the language models behave differently (i.e., memorize or generalize)?

---

> ### Author Response · Authors · 2024-11-24
> **Rebuttal to Reviewer's Comments - 6PR6**
>
> We appreciate the reviewer’s thoughtful feedback, which prompted us to conduct additional experiments, yielding encouraging initial findings that further support our hypothesis. While we acknowledge that applying this approach to broader real-world contexts (e.g., larger models, diverse tasks) requires further exploration and will be addressed in future work, we believe the current study’s contributions are significant. Sharing these findings now can provide valuable insights to the community and inspire further research to advance our understanding and control of memorization and generalization behaviors in LLMs. Below, we address the concerns and questions raised.
>
> Weakness
> 1. Generality of conclusions
>
> We acknowledge that our current experiments are limited to synthetic datasets and relatively small model scales. However, our findings represent an initial step toward understanding the localization of memorization and generalization behaviors in LLMs. While these results are derived in a controlled setting, we believe they provide a solid foundation for future exploration in more complex, real-world scenarios. In fact, recently we have found positive findings on the pretrained LLM such as Llama3-instruct, and also shared neurons for memorization/generalization across multi-tasks, and we plan to report it in the future work with more comprehensive analysis. However, we think sharing current findings now can provide valuable insights to the community and inspire more research directions.
>
> 2. Technical contributions
>
> We agree that some techniques, such as inference-time intervention (ITI) and causal analysis, build upon prior works. However, our primary contribution lies in their novel application to investigate a previously unexplored research question: understanding the interplay between neuron-level behaviors and **memorization/generalization** in LLMs. We believe this unique perspective adds value to the field and provides a starting point for deeper investigation.
>
> Questions
> 1. training mechanism
>
> We tried both fine-tuned and from-scratch mechanism. For in-context inference, we fine-tuned the pretrained GPT-2-Medium with our designed datasets, and for arithmetic addition, we trained the GPT2 model architecture from scratch. The experimental results reveal memorization/generalization behavior differentiation in both scenarios.
>
> 2.  re-phrase the input prompts still introduce additional variables
>
> The reviewer's concern is valid and appreciated. However, in order to examine the differences in the model’s internal representations between memorization and generalization behaviors, we need to ensure that the model’s internal states differ between the two behaviors (with the same input, the model internal representations are identical). To achieve this, we set the temperature to 0 to generate deterministic outputs. And the variability in behavior and internal representations is then introduced through the re-phrasing mechanism. While we acknowledge that re-phrasing could introduce additional variables, we believe these effects are minimal, allowing us to isolate the specific behaviors of memorization and generalization.

---

> ### Author Response · Authors · 2024-11-25
> **Request for Feedback on Author Response**
>
> Dear Reviewer 6PR6,
>
> Thank you for your valuable feedback. We have carefully addressed your comments in our response and would greatly appreciate it if you could kindly review it and share any further thoughts.
>
> Thank you for your time and consideration.

---

### Official Review · Reviewer_PREZ · 2024-11-03

**Soundness:** 2
**Presentation:** 2
**Contribution:** 1
**Rating:** 3
**Confidence:** 4

**Summary:**

This paper uses two synthetic dataset to detect whether a model engages in memorization or generalization behaviors. The analysis is done in three stages: (1) determining if neurons in LLMs differentiate spatially for memorization versus generalization, (2) predicting these behaviors based on hidden states, and (3) influencing LLMs to switch between these behaviors through real-time interventions. They find that deep layers are responsible for the two distinct behaviors and show that model behaviors can be controlled.

**Strengths:**

1. The paper aims to address an important question — whether the model engages in memorization or generalization. The question is of great importance for developing trustworthy AI and has broad applications, such as privacy-sensitive LLMs.

2. The analysis goes beyond simply “passively” detecting distinct neuron activation patterns; it also includes “actively” steering the model toward specific behaviors. By combining these approaches, the authors effectively differentiate the mechanisms LLMs use when engaging in memorization versus generalization, offering a more comprehensive understanding of these behaviors.

**Weaknesses:**

1. The study primarily focuses on relatively small models (GPT-2 and GPT-2-medium) and small synthetic datasets. For a paper aiming to establish an empirical pattern, a more comprehensive analysis across a range of model sizes and a broader set of tasks—ideally including less synthetic, real-world tasks—would strengthen the findings and increase confidence in the generalizability of the results.

2. The authors designed two synthetic datasets to observe distinct behaviors in models, serving as indicators of memorization versus generalization. However, the patterns identified appear limited to these specific datasets. To effectively demonstrate (a) whether LLMs exhibit spatial differentiation of neurons for memorization and generalization, and (b) the predictability of these behaviors based on internal representations, it would be important to show that the observed spatial differentiation generalizes across tasks (pattern in task A generalize to task B). From the current experiments on two tasks with two different models, the main unifying insight seems to be that behavior differentiation occurs in the later layers, which may not fully establish the robustness of the findings across varied contexts.

3. The method of detecting differences in hidden layer activations is relatively straightforward, and as noted in the first point, the findings may be limited in their novelty and broader applicability. It remains unclear whether the paper offers substantial new insights or practical implications at scale for advancing our understanding or control of LLM behaviors.

**Questions:**

Could you clarify the experimental procedure? In particular, I am confused about:

1. “The training process involved presenting each instance in its entirety to the LLM rather than as a QA pair. “ Are you finetuning with a causal language modeling objective? Could you give some concrete examples?

 2. “During training, we continuously monitored the model’s ability to perform memorization and generalization on a test set and preserved the model once it demonstrated both behaviors. “ Could you explain why you use this behaviors instead of (1) train until convergence (2) train until performance has reached a maximum on a validation set. The current choice seems an unnatural one compared to the usual practice.

---

> ### Author Response · Authors · 2024-11-24
> **Rebuttal to Reviewer's Comments - PREZ**
>
> We appreciate the reviewer’s thoughtful feedback, which prompted us to conduct additional experiments, yielding encouraging initial findings that further support our hypothesis. While we acknowledge that applying this approach to broader real-world contexts (e.g., larger models, diverse tasks) requires further exploration and will be addressed in future work, we believe the current study’s contributions are significant. Sharing these findings now can provide valuable insights to the community and inspire further research to advance our understanding and control of memorization and generalization behaviors in LLMs.
> Below, we address the concerns and questions raised.
>
> Weaknesses
> 1. Focus on Small Models and Synthetic Datasets:
>
> We fully acknowledge the limitations of our study's scope, focusing on smaller models (GPT-2 and GPT-2-medium) and synthetic datasets. These choices were intentional to maintain experimental control and clarity in identifying neuron-level behavior patterns. However, we agree that expanding the analysis to larger models and more complex, real-world tasks would provide stronger empirical support and improve the generalizability of our findings. In fact, we are actively working on experiments involving larger models (e.g., ＬLlama3) and broader tasks, and currently with some positive findings, which we hope to report in follow-up work with comprehensive analysis. However, we still think it beneficial to share the current work with the community and potentially inspire other researchers.
>
> 2. Generalization of Observed Patterns Across Tasks:
>
> While our current study demonstrates behavior differentiation primarily in two tasks, we agree that further validation across a wider range of tasks is necessary to establish the robustness of spatial differentiation and predictability of behaviors. Our current findings suggest that behavior differentiation is more pronounced in the later layers, but we acknowledge this is only a starting point. In future iterations, we plan to explore whether these patterns generalize across diverse datasets and task domains, ensuring a more comprehensive understanding. In fact, we have found that common neurons for generalization/memorization are shared among multiple tasks, which we plan to report in future work with more comprehensive analysis.
>
> 3. Novelty and Practical Implications at Scale:
>
> We appreciate the reviewer’s suggestion to clarify the broader implications of our findings. While the method for detecting hidden layer activations is relatively straightforward, the novelty lies in the combination of analysis and intervention techniques to steer memorization/generalization behaviors. The practical implications are significant, particularly for tasks requiring control over memorization and generalization, such as privacy-preserving language generation. In the revised version, we will explicitly connect our findings to potential real-world applications to better highlight their impact.
>
> Questions
>
> 1. Training Objective and Examples:
>
> Yes, we fine-tuned the models using a causal language modeling objective. For example, in one synthetic dataset, the input might be:
> "Input: 2+3+5\nTarget: 10\n"
> This format allows the model to infer either the memorized mapping (e.g., for seen inputs) or generalize (e.g., for unseen inputs) depending on the training setup. We will include additional concrete examples in the revision to clarify this process.
>
> 2. Monitoring Memorization and Generalization:
>
> We chose to monitor behaviors during training and preserve the model when it exhibited both memorization and generalization on a held-out test set because these behaviors form the focus of our study. Training to convergence or optimizing validation set performance may result in models biased toward one behavior (either memorization or generalization), which would undermine the balanced analysis we aimed to achieve. While this approach deviates from standard practices, it was specifically designed to align with our study’s objectives. We will provide a more detailed rationale for this choice in the revised manuscript.

---

> ### Author Response · Authors · 2024-11-25
> **Request for Feedback on Author Response**
>
> Dear Reviewer PREZ,
>
> Thank you for your valuable feedback. We have carefully addressed your comments in our response and would greatly appreciate it if you could kindly review it and share any further thoughts.
>
> Thank you for your time and consideration.

---

### Official Review · Reviewer_ct23 · 2024-11-03

**Soundness:** 3
**Presentation:** 3
**Contribution:** 2
**Rating:** 5
**Confidence:** 3

**Summary:**

This paper explores memorization and generalization in large language models (LLMs), inspired by the functional specialization observed in the human brain. The authors want to determine whether LLMs show differentiation among neurons when performing memorization or generalization, then use techniques to predict which behavior the model is likely to perform based on activations. And, lastly, implement inference-time interventions to direct models towards memorization or generalization as needed.

**Strengths:**

- The paper tackles a highly relevant problem in the field of LLMs—understanding and controlling the behaviors of memorization and generalization.
- The paper is clearly written, with a logical flow. The research questions and objectives are well-defined, making the study’s purpose and approach easy to follow.
- The authors construct specialized datasets that distinguish between memorization and generalization. This setup provides a good base for analysis

**Weaknesses:**

- The hypothesis that certain neurons control specific behaviors based on brain function?  While inspired by brain functionality, the paper doesn’t fully substantiate or utilize the correlation to neuroscience.

- The method relies on having a specialized dataset to differentiate between behaviors, which may limit practical applicability. Furthermore, the study only considers a single task; in real-world applications, models are often fine-tuned across multiple tasks, which may affect behavior control.

- Is the behavior shift strictly binary, meaning that applying a shift immediately moves the model to the other state (e.g., from memorization to generalization)?

- The focus on neuron-level interventions may be too granular. As this requires identification of neuron behavior using custom datasets. Exploring higher-level interventions, such as prompting or input changes to toggle memorization or generalization, might be more relevant? Thoughts?

**Questions:**

- As selecting appropriate values for topN and alpha is crucial for achieving the desired behavior shift, how can this be optimally chosen for different archs, datasets and tasks? This can also be a drawback?
- For the intervention, is the adjustment applied to all neurons or limited to those in the final layers? Visualizing the number of neurons impacted and identifying the specific layers they belong to would provide valuable insights
- Also could you please explain what is meant by "spatial" differentiation in this context

---

> ### Author Response · Authors · 2024-11-24
> **Rebuttal to Reviewer Comments**
>
> We appreciate the reviewer’s thoughtful feedback, which prompted us to conduct additional experiments, yielding encouraging initial findings that further support our hypothesis. While we acknowledge that applying this approach to broader real-world contexts (e.g., larger models, diverse tasks) requires further exploration and will be addressed in future work, we believe the current study’s contributions are significant. Sharing these findings now can provide valuable insights to the community and inspire further research to advance our understanding and control of memorization and generalization behaviors in LLMs.
> Below, we address the concerns and questions raised.
>
> Weaknesses
> 1. Correlation to Neuroscience
>
> Our inspiration stems from foundational work like Brodmann's cortical localization (Garey, 1999) and early computational models such as the perceptron (Rosenblatt, 1958), which established the link between neural and artificial systems. Recent studies further explore this connection, investigating alignment between LLMs and human brain activity (e.g., Tang et al., 2023; Aw et al., 2023; Ren et al., 2024) and drawing on neuroscience-inspired methods like neural cluster knockout (Bhile & Maes, 2024). Given this context, we believe it is intuitive and valuable to explore whether localization principles in the brain extend to LLMs, as our study contributes to this growing interdisciplinary dialogue. We will add more context regarding to the correlation to neuroscience.
>
> 2. Dataset specialization and single-task limitation
>
> We recognize that reliance on specialized datasets may limit immediate practical applicability. However, these datasets are designed to provide a controlled environment for isolating and studying behaviors, a necessary step before extending to real-world, multi-task settings. As noted in the paper, we are actively exploring how to generalize these findings to larger models and multiple tasks, for example, we further found that common neurons for generalization/memorization are shared among multiple tasks, which we plan to report in future work with comprehensive analysis.
>
> 3. Binary behavior shift
>
> Based on our experimental results, we have observed that the difficulty of transitioning from memorization to generalization differs from the reverse, suggesting that these are not strictly binary scenarios. We will clarify this observation in the paper and highlight it as an area for future exploration, where we plan to conduct more comprehensive experiments to further investigate this phenomenon.
>
> 4. Granularity of neuron-level interventions
>
> We agree that higher-level interventions, such as prompt engineering or input modifications, may complement neuron-level adjustments. However, our work focuses on neuron-level interventions as a first step to empirically identify the mechanistic underpinnings of behavior in LLMs. This granularity allows precise control and a better understanding of the internal workings, which can inform future higher-level approaches.
>
> Questions
> 1. Choosing values for topN and alpha
>
> Selecting optimal values for topN and alpha is indeed crucial. In our experiments, we empirically determine these values through grid search on the validation set. We acknowledge that optimal values may vary across architectures, datasets, and tasks. We will emphasize this in our revision.
>
> 2. Scope of Neuron Adjustments
>
> For the current study, adjustments were applied to neurons across all layers. However, the impacted neurons are targeted according to the correlation with behavior shifts, and our analysis shows that neurons in the later layers exhibit the most pronounced correlation with behavior shifts. We will include visualizations in the revised paper to highlight the distribution of impacted neurons and the specific layers they belong to.
>
> 3. Definition of "Spatial" Differentiation
>
> By “spatial differentiation,” we refer to the observation that neurons associated with memorization and generalization tend to activate in distinct regions of the latent space. This term highlights the structural separation in the model’s internal representations. We will clarify this terminology in the paper for better understanding.

---

> ### Author Response · Authors · 2024-11-25
> **Request for Feedback on Author Response**
>
> Dear Reviewer ct23,
>
> Thank you for your valuable feedback. We have carefully addressed your comments in our response and would greatly appreciate it if you could kindly review it and share any further thoughts.
>
> Thank you for your time and consideration.

---

> > ### Comment · Reviewer_ct23 · 2024-11-25
> >
> > Thank you for addressing all the questions.
> > The idea is intriguing, and the approach has potential. However, my concerns remain regarding the broader applicability of the method. The reliance on specialized datasets and its current focus on single-task scenarios and the granularity of the approach, limit its generalizability.
> > As a result, I will maintain my current score.
> >
> > A quick question, "We will include visualizations in the revised paper to highlight the distribution of impacted neurons and the specific layers they belong to." Can you point me to the figure please? I could not find it.

---

> > > ### Author Response · Authors · 2024-11-27
> > >
> > > We thank reviewer for the continued engagement and thoughtful feedback. Regarding the applicability of the method, we respectfully disagree with the concern raised. While our experiments were conducted on limited datasets and scenarios, this should not diminish the value of the work. Our findings have provided inspiring insights into the dynamics of LLM memorization and generalization, which we believe are significant contributions to the field.
> > >
> > > We emphasize that even phase-specific experimental results, when they offer meaningful insights, should be considered valuable for sharing with the research community. Such work can spark new directions and advancements within the field. Moreover, as noted in our paper, these results are not inherently restricted to narrow settings; they can be expanded to more general contexts, including pretrained LLMs and multi-task scenarios, which we are actively exploring. We hope the reviewer will assess the paper based on its potential to drive progress in the research community, rather than its immediate productized applicability.
> > >
> > > Regarding the visualization of impacted neurons, we did not include it in the current paper but would incorporate it if the paper is accepted, aligning it with the overall content. For clarity, the visualization demonstrates that most of the targeted top neurons are located in the black or white regions depicted in Figures 3 and 4.

---

### Note · Authors · 2024-12-14

I have read and agree with the venue's withdrawal policy on behalf of myself and my co-authors.